# Development of a Multi-Criteria Decision-Making Approach for Evaluating the Comprehensive Application of Herbaceous Peony at Low Latitudes

**DOI:** 10.3390/ijms232214342

**Published:** 2022-11-18

**Authors:** Xiaobin Wang, Runlong Zhang, Kaijing Zhang, Lingmei Shao, Tong Xu, Xiaohua Shi, Danqing Li, Jiaping Zhang, Yiping Xia

**Affiliations:** 1Genomics and Genetic Engineering Laboratory of Ornamental Plants, Institute of Landscape Architecture, Department of Horticulture, College of Agriculture and Biotechnology, Zhejiang University, Hangzhou 310058, China; 2Zhejiang Institute of Landscape Plants and Flowers, Hangzhou 311251, China

**Keywords:** multi-criteria decision-making (MCDM), analytic hierarchy process (AHP), herbaceous peony, germplasm resources, breeding, global warming, low latitudes

## Abstract

The growing region of herbaceous peony (*Paeonia lactiflora*) has been severely constrained due to the intensification of global warming and extreme weather events, especially at low latitudes. Assessing and selecting stress-tolerant and high-quality peony germplasm is essential for maintaining the normal growth and application of peonies under adverse conditions. This study proposed a modified multi-criteria decision-making (MCDM) model for assessing peonies adapted to low-latitude climates based on our previous study. This model is low-cost, timesaving and suitable for screening the adapted peony germplasm under hot and humid climates. The evaluation was conducted through the analytic hierarchy process (AHP), three major criteria, including adaptability-related, ornamental feature-related and growth habits-related criteria, and eighteen sub-criteria were proposed and constructed in this study. The model was validated on fifteen herbaceous peonies cultivars from different latitudes. The results showed that ‘Meiju’, ‘Hang Baishao’, ‘Hongpan Tuojin’ and ‘Bo Baishao’ were assessed as Level I, which have strong growth adaptability and high ornamental values, and were recommended for promotion and application at low latitudes. The reliability and stability of the MCDM model were further confirmed by measuring the chlorophyll fluorescence of the selected adaptive cultivars ‘Meiju’ and ‘Hang Baishao’ and one maladaptive cultivar ‘Zhuguang’. This study could provide a reference for the introduction, breeding and application of perennials under everchanging unfavorable climatic conditions.

## 1. Introduction

The importance of plant breeding for sustainable development is rising rapidly due to extreme weather events and changed climates [1,2,3]. Utilizing adapted germplasms ensures a sustained yield production and minimizes the negative impacts of climate change on agriculture and landscape ecosystems [4,5]. Breeding a new adapted cultivar is a very time-consuming and tedious process [6]. However, introducing and selecting promising genotypes directly from different latitudes or wild resources could considerably shorten the process [7,8]. Introducing new crops or cultivars—in particular, the introduction between different latitudes—not only leads to the diversification of agricultural production and applications but also has positive effects on biodiversity and ecosystem services [9,10]. In addition, the selected elite germplasms could be used as parental materials to carry out further precise breeding and research work on the molecular mechanism of stress resistance [11]. However, the introduction and selection are very limited to economic plants at low latitudes.

Herbaceous peony (*Paeonia lactiflora*) is a world-renowned economic crop with high ornamental, edible, medicinal and ecological values [3,12,13]. In recent years, herbaceous peonies have gained a new reputation as high-end cut flowers. Up to now, cut peonies have been produced in over 25 countries, with primary markets in Europe, Asia and the United States. Only in Europe, trade in cut peonies has increased 50-fold in the last 30 years, from three million stems produced in the Netherlands at the end of the 1980s to about 140 million stems from 20 countries (data from: Royal Flora, Y. Kohavi). Despite its popularity in the international market, the cultivation area of the peony is gradually limited due to the global warming. At low latitudes (N 30°00′–S 30°00′ areas), the situation is even worse, and the cultivation of the peony has encountered unprecedented challenges [12,14]. High temperatures in the spring (an average of 17–27 °C), especially combined with high precipitation (an average of 127–147 mm, data from https://zh.weatherspark.com (accessed on 10 September 2021)), can cause stem bending, flower bud abortion and severe diseases (Figure 1A–H). High temperatures in the summer, which extreme temperatures usually exceed 40 °C, could lead to severe heat damage and premature withering of the aboveground parts (Figure 1I–K) [12]. Easily formed insufficient chilling requirements due to high temperatures in the autumn and winter affect the establishment and release of dormancy and subsequent normal flowering and vegetative growth (Figure 1L,M) [15,16]. These problems have directly caused the decline of the ornamental value and application of the peony under everchanging unfavorable climatic conditions [17]. Thus, the main goal for improving the herbaceous peony at low latitudes is the screening and breeding of adapted cultivars [18].

There have been many studies on the introduction, cultivation and comprehensive evaluation of the peony, but most of them were carried out at mid and high latitudes; little work has been undertaken at low latitudes [18,19]. Liu et al. (2019) carried out a comprehensive evaluation of fifteen introduced peony cultivars in the Luoyang alpine region, Henan Province, based on an evaluation model with a global weight of 0.52 for flowering-related traits [19]. Four excellent cultivars were selected by Wu et al. (2014) using the AHP method in Beijing, China, which was established with the numbers of flowers, flower diameter, stem diameter and florescence of every single flower as the four highest weighted indices [20]. Obviously, the introduction and comprehensive evaluation of the peony at these mid and high latitudes were mostly aimed at screening the peony germplasm with high ornamental value rather than for adaptive cultivation [18,19]. The few studies that are available have focused primarily on screen-adapted herbaceous peonies basically conducted at mid-latitudes [21]. We have carried out the resource evaluation of the peony at low latitudes, but due to the limited cultivation experience, the establishment of the evaluation model also focused on the selection of high ornamental resources [18]. With longer practice and scientific research work carried out at low latitudes and continuous consultation with experts and farmers in relevant fields, adaptability, especially heat resistance, has been identified as key factor affecting the cultivation of the peony [12]. Therefore, it is especially important to establish an integrated evaluation methodology with adaptability as a major consideration to screen elite peony germplasms for sustainable growth at low latitudes [7,22].

The development of an integrated evaluation approach frequently encounters with complex multi-criteria situations [23]. A multi-criteria decision-making (MCDM) model is proven to be one of the better tools to address such complex selection issues [24,25]. Construction of a MCDM model typically includes three steps: selection of the criteria indices, weighting of the criteria indices and multi-criteria decision analysis [7,26]. The analytic hierarchy process (AHP) is one of the best subjective weighing methods for obtaining weights of each alternative in an MCDM approach, which was first established by Saaty (1980) [27,28]. The AHP approach helps decision-makers to convert subjective evaluation into objective measures, increasing the validity, efficiency and credibility of the results. It also allows for the combination of qualitative and quantitative factors in the total evaluation [29]. In recent years, MCDM has found its grounding application in various fields and disciplines, such as astronomy, environmental ecology, energy science and artificial intelligence [30,31,32,33,34]. In the agricultural context, MCDM model has been applied to a variety of crops and economic plants [35,36,37,38]. In miscanthus (*Miscanthus* spp.), the MCDM model was applied to select high-biomass and high-quality miscanthus varieties for bioenergy production [7]. Similarly, the model has also been used to select the most suitable table grape variety intended for organic viticulture [39]. Continuing under the scope of agriculture, only few studies concerning the selection of species or cultivars for adaptive cultivation under MCDM strategies [40]. Moreover, methods for selecting germplasms have primarily focused on food crops with little to no emphasis on the ornamental crops, such as herbaceous peony [18].

This study creates a multi-criterion integrated decision support framework for selecting an elite herbaceous peony germplasm at low latitudes. The MCDM model established in this study is an extension and improvement of our previous study, aiming to evaluate the comprehensive application of herbaceous peony at low latitudes [18]. The model reconstructed the AHP system, increased adaptability and growth habits-related indices, while reduced reproductive traits and ornamental values-related indices. In addition, the weight of adaptability-related indices was improved via a pairwise comparison. Finally, the model could accurately screen adaptive peony germplasm at low-latitudes and shorten the screening time from six years to three years. The establishment of this model fills a gap in the screening of adapted peonies and greatly facilitates the selection of elite germplasm at low latitudes

The paper structure is organized as follows. Section 2 shows the results associated with the MCDM model, providing an elaborate display of the case study. Section 3 discusses the application of the model and future perspectives. Section 4 describes the methodology, presenting the steps of the development and the application of the MCDM model. The objectives of this research were to: (1) develop a modified MCDM model for evaluating the comprehensive application of herbaceous peony at low latitudes and (2) select representative peony cultivars with strong adaptability for future crossbreeding and studies on the mechanism of stress resistance. This study could promote the cultivation and application of peony at low latitudes and provide a reference for the adaptive cultivation of perennial plants in the context of global warming.

## 2. Results

### 2.1. Local Weights of Criteria and Sub-Criteria Indices

Local weights of each criterion were calculated based on expert scoring (Table 1). Consistency index (CI) and consistency ratio (CR) values for each matrix were obtained and all of the CR values < 0.1 (Table 1), which indicated that the consistency of judgment matrix was acceptable.

Global weights of criteria and sub-criteria were calculated by using Equation (8) (Table 2). Generally, adaptability was the most important criteria, up to 54%, followed by ornamental features criteria (23%) and growth habits criteria, which is the least important one (16%) (Table 2).

The heat damage level (C1), root rot rate (C2), disease rate (C3), flower number per plant (C5) and group blooming period (C6) were the top five sub-criteria indices of global weight in the whole MCDM model. However, the top three sub-criteria in the global weight ranking were all adaptability-related indices (Table 2). Among the ornamental features-related criteria, the flower number per plant (C5) and group blooming period (C6) were the two sub-criteria with the largest proportion, both of which were 0.078 (Table 2). In growth habits-related sub-criteria, stem bending degree (C12) and dates of bud break (C17) were the most two crucial indices, which were 0.048 and 0.029, respectively (Table 2).

### 2.2. Observations of Adaptability-Related Traits

We have carried out the introduction, cultivation and breeding of herbaceous peony at low latitudes since 2012 [18]. High temperature and high humidity were found the most important factors restricting the popularization and application of peony under relatively high temperatures at low latitudes (Figure 1). In this study, we observed four adaptability-related sub-indices: heat damage level, root rot rate, disease rate and survival rate. ‘Hang Baishao’, ‘Meiju’, ‘Bo Baishao’ and ‘Hongpan Tuojing’ performed extremely well in these four sub-criteria indices. Specifically, these four cultivars both showed low disease rates, root rot rates and heat damage levels and maintained almost 100% survival rates after three years of cultivation (Figure 2). On the contrary, some cultivars, such as ‘Zhuguang’, ‘Taohua Feixue’ and ‘Yangfei Chuyu’ showed obvious unsuitability after introduced at low latitudes (Figure 2). These three cultivars observed gradually increased disease rates, even close to 100%, and severe heat damage and root rot, ultimately resulted in a low survival rate (Figure 2). However, most cultivars, such as ‘Zaohong’, ‘Yanzi Xiangyang’, ‘Zifeng Chaoyang’, ‘Qing Yunhong’ and ‘Shanhe Hong’ showed moderate performances in these four adaptability-related traits, and the final survival rate was between 50% and 70% after three years of cultivation. In addition, although some of the cultivars have excellent performances in some aspects of the adaptability, it is hard to combine various resistances, such as ‘Qihua Lushuang’ and ‘Chishao’, which have strong resistance to disease and root rot but lack heat resistance (Figure 2).

### 2.3. Observations of Ornamental Features-Related Traits

The ornamental features-related traits of fifteen herbaceous peony cultivars were shown in Figure 3. ‘Bo Baishao’, ‘Qihua Lushuang’, ‘Liantai’ and ‘Chishao’ performed well in several important ornamental features-related traits, such as flower number per plant, proportion of flowering plant and blooming period per flower, while ‘Zaohong’, ‘Yangfei Chuyu’ and ‘Zhuguang’ performed poorly in these flowering indices (Figure 3A–C). All cultivars showed little difference in the flower diameter indices, within the range of 10–15 cm. Notably, the flower diameters of ‘Hangbaishao’ and ‘Meiju’ decreased obviously after three years (Figure 3D). ‘Qihua Lushuang’ had the highest number of aborted flowers per plant in 2018, but sharply reduced by 2021, contrary to the performance of ‘Hang Baishao’ (Figure 3E). ‘Zaohong’ and Chishao’ bloom early and have a long group blooming period, while ‘Taohua Feixue’ and ‘Shanhe Hong’ have a late and short group flowering period (Figure 3F).

### 2.4. Observations of Growth Habits-Related Traits

‘Bo Baishao’, ‘Qihua Lushuang’, ‘Hongpan Tuojing’ and ‘Hang Baishao’ had relatively high plant heights and widths, while ‘Taohua Feixue’, ‘Yangfei Chuyu’ and ‘Zhuguang’ were cultivars with very limited plant height and width (Figure 4A,B). The plant heights and widths decreased in most cultivars, while increased in ‘Yanzi Xiangyan’, ‘Zifeng Chaoyang’, ‘Zaohong’ and ‘Hongpan Tuojing’ from 2018 to 2021. Besides, the plant heights of ‘Shanhe Hong’ and ‘Qihua Lushuang’ increased, but the plant width declined after three years (Figure 4A,B). Similarly, stem numbers of most cultivars decreased after three years, on the contrary, stem diameter of most cultivars increased (Figure 4C,D). The cultivars showed differences in the stem bending degree indices—in particular, ‘Hang Baishao’, ‘Meiju’, ‘Hongpan Tuojing’ and ‘Chishao’ performed extremely low degrees of stem bending. However, the rest of the cultivars performed poorly in this crucial index, with stem bending exceeding 20 degrees, and some cultivars, such as ‘Yanzi Xiangyan’ and ‘Qihua Lushuang’, even exceeded 40 degrees (Figure 4E). In the observation of bud break, most cultivars bud burst in the middle and early March, while a few cultivars, such as ‘Zaohong’, ‘Yanzi Xiangyang’ and ‘Zifen Chaoyang’, sprouted in the middle and late February (Figure 4F). In addition, most cultivars had chlorophyll content between 50–60 during full blooming, except for ‘Zaohong’, ‘Shanhe Hong’ and ‘Liantai’ (Figure 4G).

### 2.5. Evaluation of Comprehensive Performance of the Fifteen Cultivars by the MCDM Model

Details in comprehensive scores and levels of the fifteen cultivars were presented in Table 3. ‘Meiju’ acquired the highest points 89.56, and ‘Hang Baishao’ was ranked second, with 85.05 points, followed by ‘Hongpan Tuojing’ and ‘Bo Baishao’, which were the four cultivars with over 80 points. These four cultivars with more than 80 points were classified as level I, indicating they have excellent comprehensive performance and are recommended as germplasms for cultivation and application at low latitudes. The comprehensive scores of ‘Qihua Lushuang’, ‘Liantai’, ‘Chishao’, ‘Qing Yunhong’, ‘Zifeng Chaoyang’ and ‘Zaohong’ were between 60–80 points and classified as level II, indicating that their comprehensive performance is ordinary and could be used as alternative application materials. At the bottom of the ranking, ‘Yanzi Xiangyang’, ‘Shanhe Hong’, ‘Taohua Feixue’, ‘Yangfei Chuyu’ and ‘Zhuguang’ were listed. These five cultivars were scored below 60 points and were classified as level III, not recommended for use at low latitudes (Table 3).

### 2.6. Chlorophyll Fluorescence of the Selected Adaptive and Maladapted Cultivars

‘Meiju’ and ‘Hang Baishao’ were screened two adaptive cultivars while ‘Zhuguang’ was the maladapted cultivar based on the comprehensive evaluation. Chlorophyll fluorescence of the three cultivars was detected in the summer of 2021. The maximum quantum yield of photosystem II (Fv/Fm), nonphotochemical quenching (NPQ), quantum efficiency of photosystem II (YII), photochemical quenching coefficient (qP) values of the three cultivars overall decreased first and then stabilized while apparent electron transfer rate (ETR) behaves in the opposite under high summer temperatures (Figure 5). Among three cultivars, ‘Meiju’ produced significantly the highest values of the four chlorophyll fluorescence indicators, followed by ‘Hangbaishao’ and the lowest values were recorded in ‘Zhuguang’ and were correlated with the chlorophyll fluorescence colors (Figure 5). Additionally, the chlorophyll fluorescence values of the cultivars were consistent with the comprehensive scores calculated by the MCDM model (Table 3).

## 3. Discussion

### 3.1. The Necessity of Introduction and Selection of Adaptive Peony Germplasm at Low Latitudes

Herbaceous peony is a famous ornamental crop worldwide, with the widespread popularity of cut flowers in recent years [17,41]. However, peony is often subjected to multiple abiotic stresses of high temperature and humidity, which seriously and initially affect its normal growth and subsequent flowering under environmental conditions at low latitudes [42,43]. Global warming exacerbates the problem, and extreme temperatures at low latitudes commonly exceed 40 °C in the summer [2,12]. Thus, evaluating germplasm resources, particularly for the purpose of selecting resistant species, is a fundamental work of peony breeding [18]. Numerous introduction and evaluation studies have been carried out in a variety of food crops, but few in ornamental crops, such as herbaceous peony [44]. In fact, herbaceous peony is an ornamental perennial with strong adaptability and could widely distributed in temperate regions of the Northern Hemisphere [13]. Moreover, there are plentiful wild resources, medicinal and ornamental species or cultivars of peonies (both herbaceous peony and tree peony) existed in the mid-latitude region of China, it is completely achievable to directly select excellent existing peony germplasms based on introduction and resource evaluation at low latitudes [18,45]. Therefore, screening or breeding adapted peony germplasm has important theoretical and practical significance in low-latitude areas, especially in the context of global warming [12,42].

### 3.2. The Development of a Specific Objective-Based Comprehensive Evaluation Model

All models are not omnipotent, their accuracy depends on the specific application purpose [46]. Some common steps in the construction of a completed MCDM evaluation model include: selecting indices, creating criteria and assigning weights (Figure 6). First and foremost, these steps should be completed in accordance with the specific purpose and characteristics of the target plant [18]. When selecting energy-related agronomic traits, dry matter yield is the primary index for screening elite germplasms [7]; when evaluating the rapeseed varieties, economic criterion is the most important index [37]; when establishing a model for selecting herbaceous peony species under protective cultivation conditions, ornamental characteristics are almost all considerations [37]. In this study, the MCDM model was aimed at solving the problem of adaptability in the process of introduction and cultivation of peony at low latitudes and, secondly, considering ornamental values and growth habits. As a consequence, the weights of the three criterion layers of the MCDM model were: adaptability (B1) > ornamental features (B2) > growth habits (B3), and the adaptability (B1) criteria had the highest weight in the criterion layer, up to 0.54 (Table 2). Additionally, the heat damage level (C1), root rot rate (C2) and disease rate (C3) were the top three evaluation sub-criteria of the MCDM model (Table 2), which is in line with our expectations. Specifically, these three adaptability-related sub-criteria have been widely used in studies of peony, rapeseed and grape [19,37,39]. In addition, these sub-criteria could typically reflect the stress resistance to high temperature and high humidity environments based on our cultivation practice at low latitudes. Therefore, the evaluation model established in this study was in accordance with our research purpose.

### 3.3. Selection of Elite Peony Germplasms at Low Latitudes

‘Meiju’, ‘Hang Baishao’, ‘Hongpan Tuojing’ and ‘Bo Baishao’ were the four cultivars that scored over 80 points and classified as level I by the MCDM model (Table 3). These four cultivars both have excellent comprehensive performances in adaptability, ornamental features and growth habits, especially in terms of strong heat and humidity resistance, recommended as elite germplasms for cultivation and application at low latitudes (Figure 2, Figure 3 and Figure 4). These cultivars could also be valuable for breeding brilliant new germplasms with strong stress resistance [47,48]. In terms of this study, these four cultivars and ‘Zhuguang’, a representative unsuitable cultivar, could be used as contrasting plant materials to determine the mechanisms governing the differences in the mechanism of stress tolerance in peony [48]. Notably, Liu et al. screened ‘Taohua Feixue’ and ‘Yangfei Chuyu’ as excellent cultivars for planting in Guanzhong area of Shaanxi Province by AHP method in 2013, while in this study, these two cultivars were scored below 60 points [21]. This indicates that the performance of the same cultivar varies greatly in different regions, further emphasizing the necessity for comprehensive evaluation of germplasm to reduce economic costs before large-scale introduction of cultivation. In addition, ‘Yanzi Xiangyang’ showed a tendency to gradually adapt at low latitudes after three years of cultivation, with fewer diseases, improved flowering number and flowering rate and increased plant height and plant width, which deserves a longer time observation (Figure 2, Figure 3 and Figure 4).

### 3.4. Accuracy and Reliability of the MCDM Model

The MCDM model constructed in this study aimed to represent an integrated evaluation research strategy based on multi-year cultivation, multi-indices observations and specific application purposes rather than a fixed formula (Figure 6 and Figure 7). The weights of sub-criteria indices will fluctuate once the tested germplasms change since the objective weight value varies with the measured data. The peonies adopted in this study were carefully selected cultivars with different performances in stress resistance and ornamental characteristics at low latitudes based on previous studies [12,18/]. In particular, the native-specific cultivar ‘Hangbaishao’, which has been proved to be an elite germplasm at low latitudes, was added as a reference for comparing selection results [12,18,49]. In the evaluation of this study, ‘Hang Baishao’ still performed excellently and ranked second overall (Table 3), which on the one hand showed the rationality and accuracy of this model, on the other hand, it re-emphasized the value of this native germplasm.

Chlorophyll fluorescence reflects photosynthetic performance and stress in algae and plants, is now widespread in various studies [50,51,52]. Sharma et al. have successfully identified wheat (*Triticum aestivum* L.) cultivars in tolerance to heat stress by using Fv/Fm [53]. In addition to screening for heat-tolerant wheat, chlorophyll fluorescence has also been used as a method for screening potential wheat cultivars adapted to water deficit environments [54]. In the herbaceous peony, the chlorophyll fluorescence intensity was consistent with the results of comprehensive heat tolerance assessment of different peony cultivars [12]. Thus, in this study, the model was further validated in identified cultivars ‘Meiju’, ‘Hang Baishao’ and ‘Zhuguang’ by the detection of chlorophyll fluorescence. The result showed that the chlorophyll fluorescence values of the cultivars were consistent with the comprehensive scores calculated by the MCDM model (Figure 5 and Table 3). Furthermore, we have previously conducted principal component analysis (PCA) and subordinate function value analysis on six peony cultivars introduced from different latitudes, ‘Hang Baishao’ and ‘Meiju’ were determined to be the most heat-tolerant cultivars, while ‘Zhuguang’ was determined to be the most heat-sensitive one of the six cultivars [12]. Although different evaluation methods were used, the evaluation results were highly consistent, which further demonstrated the reliability and stability of the MCDM model established in this study.

### 3.5. Limitations, Recommendations and Future Perspectives

The MCDM model is low-cost, timesaving and suitable for screening adapted peony germplasm under hot and humid climates. Fifteen tuberous roots and nine square meters of land surface are sufficient for screening one cultivar. By using this model, it takes only three years to screen the expected cultivars, which is three years less than using our previous model. The model proposed in this study can be directly used at low latitudes, specifically, substituting the observational data of the tested cultivar into the model, then calculate the comprehensive evaluation points, and if it is greater than 80 points, the cultivar is selected. This model requires certain modifications for use in other perennials or geophytes, in that case, due to the specific characteristics of each plant, the major importance is to consider local experts’ knowledge and experience, which would possibly add, remove or modify important criteria. Nevertheless, we do not recommend its use in cases where there are large differences in climate and species, as the present model involves a limited number of criteria that may require fundamental modifications.

The model is also a continuous improvement model that will advance in the future. It is well known that with the increasing global warming, the growing environment of plants may face more severe deterioration in the future. This model will be modified accordingly, for example, adding indices such as heat stress oxidation and high light damage, or changing the weights of some indices. In addition to the model itself, this study also aims to provide a research strategy (Figure 6) for other plants with similar situations as peony.

## 4. Materials and Methods

### 4.1. Development of a MCDM Model

Figure 6 structurally displayed the main process of developing a complete MCDM model of herbaceous peony in this study. The criteria indices in the current study were selected based on information from the literature and multi-years practice of cultivation (Section 4.1.1). The weights of these criteria and sub-criteria were calculated by the AHP method (Section 4.1.2), and the elite cultivars were selected via the final comprehensive evaluation (Section 4.1.3). The detailed steps are described in the following sections:

#### 4.1.1. Identify Relevant Criteria Indices

The criteria indices were selected through the following two steps in current study: (1) The literature on traits related to comprehensive evaluation of peony cultivation and application, especially when introduced from regions of different latitudes and encountered unfavorable climatic conditions. All reported traits were compared and selected after preliminary screening; (2) Identifying the main limiting factors of peony through years of practice of cultivation at low latitudes. High temperature and high humidity were found the most important factors restricting the popularization and application of peony through years of practice of cultivation at low latitudes (Figure 1). Considering the two main limiting factors, heat damage level, root rot rate, disease rate and survival rate were identified as the corresponding adaptability indices, which have been proven in our previous studies to respond well to the resistance of peonies to humid and hot environments. Based on the above, the core traits with high recommendations were selected for the model establishment (Table 4).

#### 4.1.2. Weighting the Criteria Indices

AHP is one of the most common subjective methods and was adopted to weight the criteria indices in the present study [56]. Its weighting procedures are described as follows [57]:Step 1: Building the AHP system

The AHP system is usually defined as a tree, where the main objective is the target layer (A), like the top of the tree; the criteria indices are the second layer (B) for evaluating the target layer, like the trunk; the third layer is the specific sub-criteria (C), and the alternatives are the roots [46,58]. In present study, the AHP system was built in four levels from top to bottom (Figure 7). The selection of adapted peony cultivars at low latitudes (A) is the main objective. The adaptability-related (B1), ornamental features-related (B2) and growth habits-related (B3) traits were selected as the three criteria indices. The eighteen specific traits belonging to the three main criteria indices are: heat damage level (C1), root rot rate (C2), disease rate (C3), survival rate (C4), flower number per plant (C5), group blooming period (C6), flower type (C7), proportion of flowering plant (C8), aborted flowers per plant (C9), blooming period per flower (C10), flower diameter (C11), plant height (C12), plant width (C13), stem number (C14), stem diameter (C15), stem bending degree (C16), dates of bud break (C17) and chlorophyll content (C18), respectively (Figure 7). The fifteen peony cultivars are the alternatives. Observations and measurements of these traits (sub-criteria indices) and their references were shown in Table 4.

Step 2: Constructing the pairwise comparison matrix

In this step, the priority weights of the above-mentioned sub-criteria indices are calculated using a square matrix of pairwise comparisons, as shown in Equation (1) [59].
(1)$$A\;\left(a_{\mathrm{ij}}\right)\;\mathrm{nxn}=\left[\begin{array}{cccccc} a_{11} & a_{12} & \ldots & a_{1j} & \ldots & a_{1n}\\ a_{21} & a_{22} & \ldots & a_{2j} & \ldots & a_{2n}\\ \ldots & \ldots & \ldots & \ldots & \ldots & \ldots \\ a_{i1} & a_{i2} & \ldots & a_{\mathrm{ij}} & \ldots & a_{\mathrm{in}}\\ \ldots & \ldots & \ldots & \ldots & \ldots & \ldots \\ a_{n1} & a_{n2} & \ldots & a_{\mathrm{nj}} & \ldots & a_{\mathrm{nn}} \end{array}\right],$$
where i represents the serial number of the former traits, j represents the serial number of the latter traits and n represents the total number of traits in matrix A. a_ij_ is the ratio of the importance of the ith trait compared with the jth trait, and it is scored according to Table 5 [60,61].

Survey experts and front-line workers in the related field to confirm the importance of these selected traits based on the above steps [62]. A panel of four experts and two workers was contacted and asked about the quantification of the importance of the criteria included in the AHP system (Figure 7). The experts responded to a special questionnaire created for this study, an example of the questionnaire was shown in Table 6.

The pairwise comparison matrix was then established after expert judgment in this study (Table 7).

Step 3: Normalizing the pairwise comparison matrix

Normalize the pairwise comparison matrix using Equation (2).
(2)$$w_{\mathrm{ij}}\prime =\frac{w_{\mathrm{ij}}}{\sum_{i=1}^{n}w_{\mathrm{ij}}}$$
where w_ij_ represents the pairwise comparison value, and w_ij_′ is the pairwise comparison value after normalization [63].

Step 4: Calculating consistency ratio of the pairwise comparison matrix

Assess the eigenvalue and the eigenvector using Equations (3)–(5):(3)$$w_{i}=\frac{\sum_{j=1}^{n}w_{\mathrm{ij}}^{\prime }}{n}$$
(4)$$W=\left[\begin{array}{c} w_{1}\\ w_{2}\\ \ldots \\ w_{i}\\ \ldots \\ w_{n} \end{array}\right]$$
(5)$$\lambda \max=\frac{1}{n}\sum_{i=1}^{n}\left({\mathrm{AW}}_{i}\right)/w_{i}$$
where w_i_ is the eigenvalue, W is the eigenvector of the matrix A and λmax is the largest eigenvalue of the pairwise comparison matrix.

Check the consistency index using Equations (6) and (7):(6)$$\mathrm{CR}=\frac{\mathrm{CI}}{\mathrm{RI}}$$
(7)$$\mathrm{CI}=\frac{\lambda \max-\;n}{n-1}$$
where n denotes the number of criteria, and CR and CI are the consistency ratio and consistency index of the pairwise comparison matrix. RI represents the random consistency index that was introduced by Saaty [58], shown in Table 8.

When CR < 0.1, the consistency degree of judgment matrix A is considered to be within the allowable range, and the eigenvectors of A could be performed to carry out the weight vector calculation [64]; however, if CR ≥ 0.1, the judgment matrix A should be considered for correction [65,66].

Step 5: Computing the global weights

The matrix A in the criteria level contains a series of criteria indices, including (a_1_,a_2_,…,a_i_,…,a_n_), their eigenvalues should be (w_1_,w_2_,…,w_i_,…,w_n_), respectively; Matrix B in the sub-criteria level belonging to a_i_ contains several sub-criteria indices, including (b_1_,b_2_,…,b_α_,…,b_β_), their eigenvalues then should be (w_1_′,w_2_′,…,w_α_′,…,w_β_′), respectively. Finally, the global weights of (b_1_,b_2_,…,b_α_,…,b_β_) should be (w_i_w_1_′,w_i_w_2_′,…,w_i_w_α_′,…,w_i_w_β_′), respectively [58].
W_g_ = w_i_ w_α_′(8)
where W_g_ is the global weight of the sub-criteria; w_i_ is the local weight of criteria and w_α_′ is the local weight of sub-criteria.

#### 4.1.3. Evaluating the Comprehensive Points of Alternatives

The rating scale of all sub-criteria was established based on the literature review, basic knowledge of the various traits of the peony and cultivation practice at low latitudes. The rating scale was divided into three levels, including highly relevant, moderately relevant and slightly relevant with scores of 1, 2/3 and 1/3, respectively (Table 9).

Evaluate the comprehensive points of alternatives by accumulating the scores of all sub-criteria (Equation (9)) [63]. The alternatives are rated according to the comprehensive evaluation score, with points between 80 and 100 as Level I; between 60 and 80 as Level II; between 0 and 60 as Level III.
(9)$$P_{\;}=100\sum_{i=1}^{n}W_{\mathrm{gi}}R_{i}$$
where P is the comprehensive evaluation points; Wg_i_ is the global weight of the i-th sub-criteria; Ri is the score of the i-th sub-criteria.

### 4.2. Application of the MCDM Model—A Case Study

To testify the feasibility of this developed model for comprehensive evaluation of herbaceous peony, fifteen representative peony germplasms (*Paeonia lactiflora*) have been selected as a case study (Figure 8). Among these fifteen cultivars, fourteen cultivars were selected midlatitude cultivars introduced from Heze City (E 34°39′-35°52′, N 114°45′-116°25′), Shandong Province, and one cultivar. ‘Hang Baishao’. was selected native low-latitude cultivar (Figure 8) [18]. ‘Hang Baishao’ is a unique traditional Chinese herbaceous peony with strong resistance to heat and humidity and low-chilling requirement trait and can be long cultivated at low latitudes in China [48,49,55]. These fifteen cultivars belong to five flower forms and have different performances in stress resistance and growth habit based on the previous studies [12,18].

In autumn of 2017, four-year-old peony tuberous roots of these fifteen cultivars were cultivated in the low-latitude Perennial Flower Resources Garden of Zhejiang University in Hangzhou (E 118°21′-120°30′, N 29°11′-30°33′), Zhejiang Province, China. Hangzhou has a subtropical monsoon climate, mild and humid. Spring is warm and humid. Summer is hot, with lots of rain and frequently occurring extreme high temperatures (Figure 9). The average annual temperature in Hangzhou is 15–17 °C. The average annual precipitation is 1100–1600 mm, and the average humidity is 76% [67]. The air temperature during the growing seasons in 2018–2021 were shown in Figure 9.

Normal field management and fertilization were carried out after introduction. Peony tuberous roots were planted in full sun with rich soil and great drainage. Fertilizer applications were conducted twice times a year, the first time was made during budding and flowering in the spring, and the second time was carried out in the fall to produce roots. Phosphate fertilizer was used in the spring, and compost was used in the fall. In the late summer, when peony leaves lose their luster, turn colors and begin to die back for the winter, cut back peony stems to three to four inches above the ground and throw away the leaves. Additionally, these peony plants were grown under natural sunlight without any shade in summer.

The seventeen criteria were observed in the field in 2018 and 2021. Fifteen plants (three biological replicates and five plants per replicate) of each cultivar were monitored, and details of the observations and measurements of evaluation indices were listed in Table 4. The data of the tested cultivars were substituted into the model to calculate the comprehensive evaluation point, and the cultivar was selected when the value was greater than 80 points. The selected brilliant cultivars will continue to be observed for a longer time to confirm whether the model is reliable.

### 4.3. Chlorophyll Fluorescence Determination of Selected Adapted and Maladapted Cultivars

Chlorophyll fluorescence is now widespread, used to monitor the photosynthetic performance of plants [52], especially the parameter Fv/Fm, which has been widely used to identify heat-tolerant genotypes as a physiological marker [50,68]. We have previously measured Chlorophyll fluorescence of peony cultivars from different latitudes and found that the parameters can accurately reflect the heat resistance of peony [12]. In this study, heat damage level is also a crucial sub-criterion for screening adaptive peony cultivars. Therefore, comparing the evaluation results obtained from the MCDM model with the observation results of Chlorophyll fluorescence indices could confirm whether the model is reliable. For validation, this study only selected the most adapted and maladapted cultivars evaluated by the MCDM model, as well as the verified reference fine peony ‘Hang Baishao’, for Chlorophyll fluorescence determination. The chlorophyll fluorescence characteristics were observed by an Imaging-PAM Chlorophyll fluorescence system (Hansatech Instruments, Norfolk, England). After a 30-min dark adaptation, the basal fluorescence (Fo) and Fv/Fm were measured on three sun-exposed, fully expanded leaves per cultivar (three biological replicates, one plant per replicate) at 10:00 a.m. on July 15 in 2021. NPQ, YII, qP and ETR parameters were also measured according to Li et al. (2021) [69].

### 4.4. Data Analysis

The experiments mentioned in this study were following a completely randomized design. Analysis of variance (ANOVA) was adopted to determine the statistical significance of the differences using SPSS 20.0 (IBM Corp., Armonk, NY, USA). Pictures or photos were combined and arranged by the Microsoft Office PowerPoint 2016. GraphPad Prism 8.0 (GraphPad Software, Inc., La Jolla, CA, USA) was used for visualization of the experimental data.

## 5. Conclusions

Establishing a comprehensive evaluation method to screen elite peony germplasm with adaptation as the main consideration is particularly important in the context of global warming. This study proposed a modified MCDM model for assessing peonies adapted to low-latitude climates, which is an extension and improvement of our previous study. The model reconstructed the AHP system, increased adaptability and growth habit-related indices while reduced reproductive traits and ornamental values-related indices. In addition, the weight of adaptability-related indices was improved via a pairwise comparison which was obtained through expert questionnaires. The model was validated on fifteen herbaceous peonies cultivars from different latitudes. The results showed that ‘Meiju’, ‘Hang Baishao’, ‘Hongpan Tuojin’ and ‘Bo Baishao’ were assessed as Level I, which have strong growth adaptability and high ornamental values, and were recommended for promotion and application at low latitudes. The reliability and stability of the MCDM model were further confirmed by measuring the Chlorophyll fluorescence of the selected adaptive and maladaptive cultivars. Consequently, the MCDM model developed in this study is low-cost, timesaving and could accurately screen adaptive peony germplasm at low-latitudes with hot and humid climates. This model fills a gap in the screening of adapted peonies and greatly facilitates the selection of elite germplasm at low latitudes. In addition to the establishment of the model; this study also provides a research strategy for other plants with similar situations as peony. In the future, the model will be modified accordingly with the increasing severity of global warming. Specifically, we could improve the applicability of the model by changing the weights of the indices, but there is no doubt that the adaptability-related index is still the most important. Finally, the application and promotion of the peony, as well as other perennial crops at low latitudes, can be addressed by policymakers supporting sustainable development and supporting land use and fundamental research funding.

## Figures and Tables

**Figure 1 ijms-23-14342-f001:**
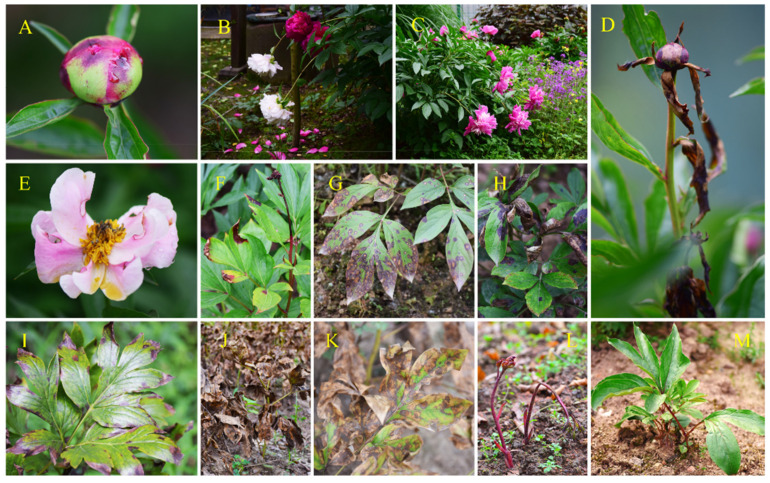
Challenges encountered in the cultivation and application of the herbaceous peony at low latitudes. (**A**–**H**) High spring temperatures and high humidity caused stem bending, flower bud abortion and severe diseases; (**I**–**K**) High summer temperatures induced heat damage and premature withering of aboveground parts; (**L**,**M**) High autumn and winter temperatures caused insufficient chilling requirement and induced subsequent abnormal flowering and vegetative growth.

**Figure 2 ijms-23-14342-f002:**
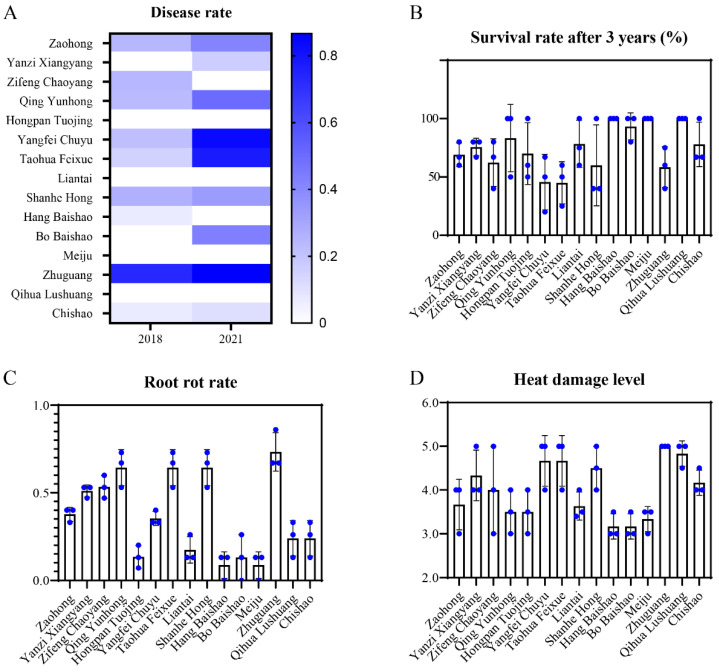
Observations and changes of adaptability-related indices of the fifteen peony cultivars. (**A**) Disease rate; (**B**) Survival rate after three years; (**C**) Root rot rate; (**D**) Heat damage level. Values represent the means± standard deviation of three replicates.

**Figure 3 ijms-23-14342-f003:**
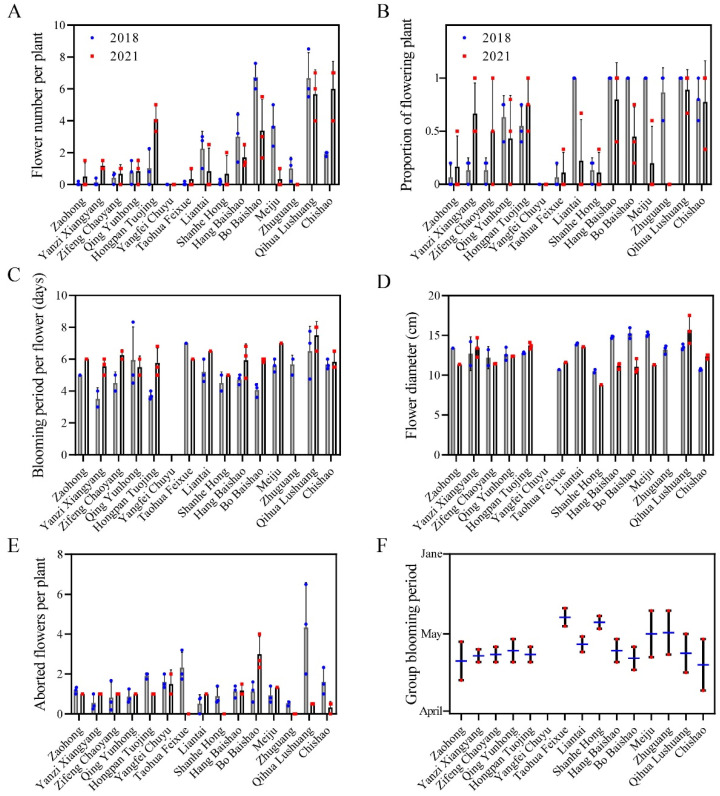
Observations and changes of ornamental features-related indices of the fifteen peony cultivars. (**A**) Flower number per plant; (**B**) Proportion of flowering plant; (**C**) Blooming period per flower; (**D**) Flower diameter; (**E**) Aborted flowers per plant; (**F**) Group blooming period. Values represent the means ± standard deviation of three replicates.

**Figure 4 ijms-23-14342-f004:**
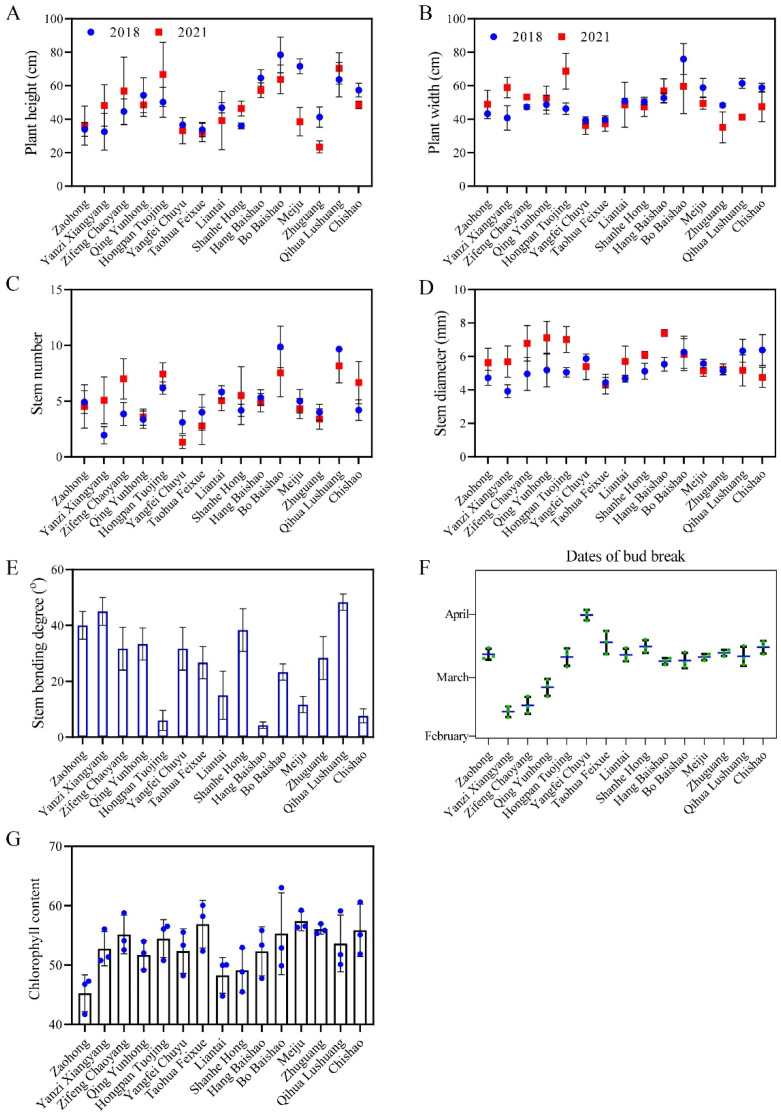
Observations and changes of growth habits-related indices of the fifteen peony cultivars. (**A**) Plant height; (**B**) Plant width; (**C**) Stem number; (**D**) Stem diameter; (**E**) Stem bending degree; (**F**) Dates of bud break; (**G**) Chlorophyll content. Values represent the means ± standard deviation of three replicates.

**Figure 5 ijms-23-14342-f005:**
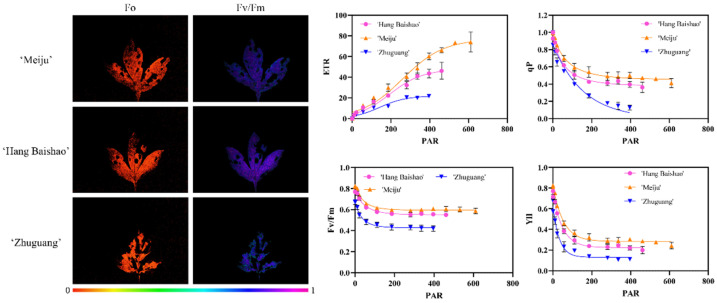
Chlorophyll fluorescence imaging screens of the selected adaptive and maladapted cultivars under summer heat stress in 2021. The fluorescence color indicates the Fo and Fv/Fm values. Data expressed in the figure represents the mean ± standard deviation of three replicates.

**Figure 6 ijms-23-14342-f006:**
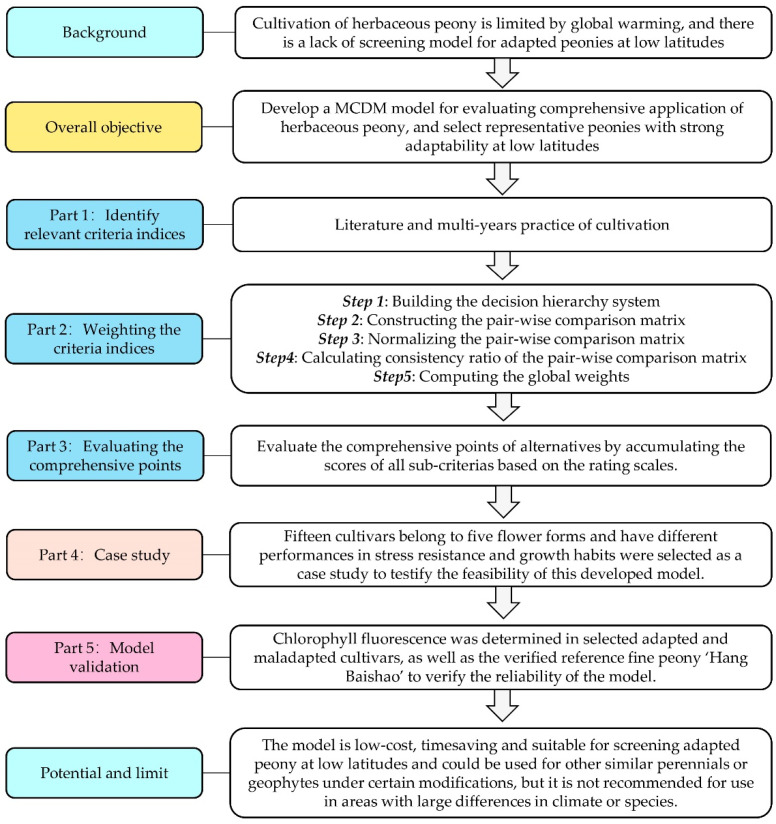
The process to create a multi-criterion integrated decision support framework for selecting elite herbaceous peony germplasm at low latitudes in this study.

**Figure 7 ijms-23-14342-f007:**
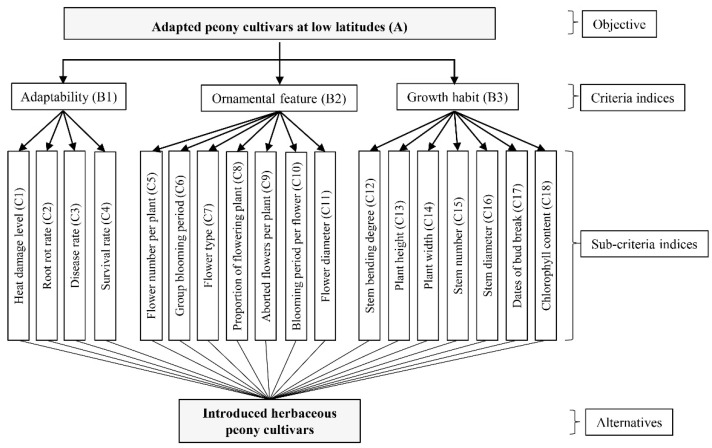
Decision hierarchy system of the MCDM model for evaluating the application potential of herbaceous peony cultivars at low latitudes.

**Figure 8 ijms-23-14342-f008:**
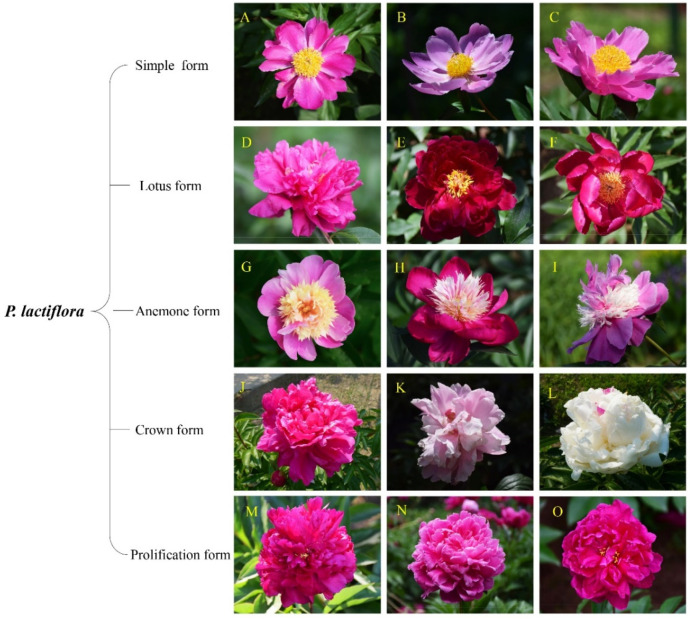
Fifteen *P. lactiflora* cultivars adopted as the material in this study. (**A**) ‘Bo Baishao’; (**B**) ‘Chishao’; (**C**) ‘Hang Baishao’; (**D**) ‘Zhuguang’; (**E**) ‘Yanzi Xiangyang’; (**F**) ‘Hongpan Tuojing’; (**G**) ‘Liantai’; (**H**) ‘Meiju’; (**I**) ‘Qihua Lushuang’; (**J**) ‘Shanhe Hong’; (**K**) ‘Taohua Feixue’; (**L**) ‘Yangfei Chuyu’; (**M**) ‘Zaohong’; (**N**) ‘Zifeng Chaoyang’; (**O**) ‘Qing Yunhong’.

**Figure 9 ijms-23-14342-f009:**
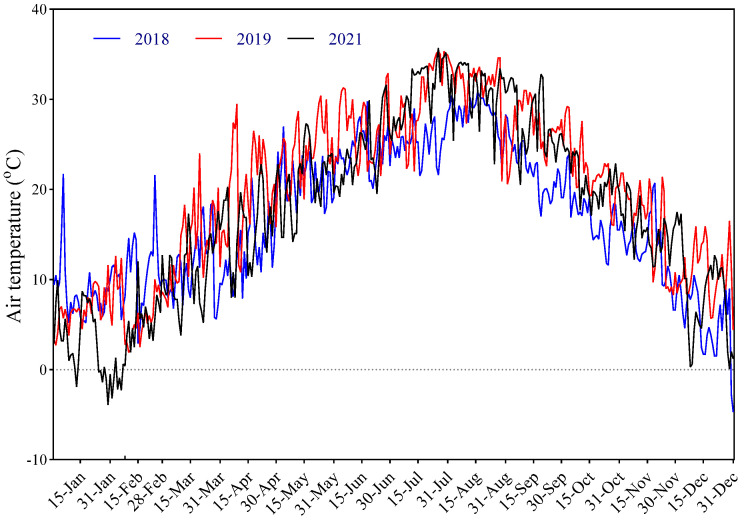
Air temperatures in the garden field in Hangzhou during the growing seasons in 2018–2021. Temperatures were recorded hourly in the field using a GM200-TH temperature and humidity recorder (Zhituo Instruments Limited Company, Hangzhou, China).

**Table 1 ijms-23-14342-t001:** The local weights of criteria and consistency test of the pairwise comparison matrix.

Matrixes	Criteria	Local Weights	Consistency Test		
			λmax	CI	CR
A-B	Adapted peony cultivars at low latitudes (A)		3.009	0.005	0.009
	Adaptability (B1)	0.540			
	Ornamental features (B2)	0.297			
	Growth habits (B3)	0.163			
B1-C	B1		4.010	0.003	0.004
	Heat damage level (C1)	0.423			
	Root rot rate (C2)	0.227			
	Disease rate (C3)	0.227			
	Survival rate (C4)	0.122			
B2-C	B2		7.346	0.058	0.042
	Flower number per plant (C5)	0.261			
	Group blooming period (C6)	0.261			
	Flower type (C7)	0.090			
	Proportion of flowering plant (C8)	0.136			
	Aborted flowers per plant (C9)	0.075			
	Blooming period per flower (C10)	0.118			
	Flower diameter (C11)	0.058			
B3-C	B3		7.037	0.006	0.005
	Stem bending degree (C12)	0.297			
	Plant height (C13)	0.097			
	Plant width (C14)	0.056			
	Stem number (C15)	0.097			
	Stem diameter (C16)	0.178			
	Dates of bud break (C17)	0.178			
	Chlorophyll content (C18)	0.097			

A is the main objective, B1–B3 are the three criteria and C1–C18 are the eighteen sub-criteria in this study; λmax is the largest eigenvalue of the pairwise comparison matrix.

**Table 2 ijms-23-14342-t002:** Global weights of the three criteria and eighteen sub-criteria.

Criteria (B)	Weights	Sub-Criteria (C)	Global Weights	Ranking
Adaptability (B1)	0.54	Heat damage level (C1)	0.228	1
		Root rot rate (C2)	0.123	2
		Disease rate (C3)	0.123	3
		Survival rate (C4)	0.066	6
Ornamental features (B2)	0.30	Flower number per plant (C5)	0.078	4
		Group blooming period (C6)	0.078	5
		Flower type (C7)	0.027	12
		Proportion of flowering plant (C8)	0.041	8
		Aborted flowers per plant (C9)	0.023	13
		Blooming period per flower (C10)	0.035	9
		Flower diameter (C11)	0.017	14
Growth habits (B3)	0.16	Stem bending degree (C12)	0.048	7
		Plant height (C13)	0.015	15
		Plant width (C14)	0.009	18
		Stem number (C15)	0.015	17
		Stem diameter (C16)	0.029	11
		Dates of bud break (C17)	0.029	10
		Chlorophyll content (C18)	0.015	16

**Table 3 ijms-23-14342-t003:** Statistics of comprehensive evaluation points and rating levels of the fifteen herbaceous peony cultivars.

Cultivars	Adaptability				Ornamental Features							Growth Habits							Points	Levels
	C1	C2	C3	C4	C5	C6	C7	C8	C9	C10	C11	C12	C13	C14	C15	C16	C17	C18		
Meiju	22.85	6.60	12.28	12.28	5.22	7.84	1.80	2.72	1.50	3.55	1.74	3.17	1.03	0.59	1.03	1.90	1.90	1.55	89.56	I
Hang Baishao	22.85	6.60	12.28	12.28	5.22	2.61	0.90	4.08	1.50	2.37	1.16	4.76	0.52	0.59	1.55	2.85	1.90	1.03	85.05	I
Hongpan Tuojing	22.85	4.40	12.28	12.28	5.22	5.22	0.90	2.72	1.50	1.18	1.74	4.76	1.03	0.59	1.55	2.85	1.90	1.03	84.02	I
Bo Baishao	15.23	6.60	12.28	12.28	7.84	7.84	0.90	4.08	0.75	2.37	1.74	3.17	0.52	0.30	1.55	2.85	1.90	1.55	83.73	I
Qihua Lushuang	7.62	6.60	12.28	12.28	7.84	7.84	1.80	4.08	0.75	3.55	1.74	1.59	0.52	0.59	1.55	1.90	1.90	1.03	75.44	II
Liantai	15.23	4.40	12.28	12.28	2.61	5.22	1.80	2.72	2.25	2.37	1.74	3.17	1.55	0.89	1.55	1.90	1.90	0.52	74.39	II
Chishao	7.62	4.40	12.28	12.28	5.22	7.84	0.90	4.08	2.25	2.37	1.16	4.76	1.03	0.59	1.55	1.90	0.95	1.55	72.73	II
Qing Yunhong	22.85	6.60	8.18	4.09	2.61	2.61	2.70	2.72	2.25	2.37	1.16	1.59	1.03	0.59	0.52	2.85	2.85	1.03	68.62	II
Zifeng Chaoyang	15.23	4.40	12.28	8.18	2.61	5.22	2.70	1.36	2.25	2.37	1.16	1.59	1.03	0.59	1.55	1.90	2.85	1.03	68.32	II
Zaohong	15.23	4.40	8.18	8.18	2.61	5.22	2.70	1.36	1.50	2.37	1.16	1.59	0.52	0.89	1.03	1.90	1.90	0.52	61.27	II
Yanzi Xiangyang	7.62	4.40	12.28	8.18	2.61	2.61	0.90	1.36	2.25	1.18	1.74	1.59	1.55	0.89	0.52	0.95	2.85	1.03	54.52	III
Shanhe Hong	7.62	4.40	8.18	4.09	2.61	2.61	1.80	1.36	2.25	1.18	0.58	1.59	1.55	0.89	1.03	1.90	1.90	0.52	46.07	III
Taohua Feixue	7.62	2.20	4.09	4.09	2.61	5.22	1.80	1.36	1.50	3.55	1.16	3.17	0.51	0.30	0.52	0.95	1.90	1.55	44.11	III
Yangfei Chuyu	7.62	2.20	4.09	8.18	2.61	2.61	1.80	1.36	1.50	2.37	1.16	1.59	0.51	0.30	0.52	1.90	0.95	1.03	42.30	III
Zhuguang	7.62	2.20	4.09	4.09	2.61	2.61	2.70	1.36	2.25	2.37	1.16	3.17	0.52	0.89	0.52	0.95	1.90	0.51	41.53	III

The full name of the sub-criteria C1–C18 were listed in Table 1.

**Table 4 ijms-23-14342-t004:** Details of evaluation indices and measurements.

Criteria	Sub-Criteria		Definitions and Measurements	Objective	References
Adaptability	Heat damage level	C1	HDL was classified into six categories according to the proportion of leaves that showed signs of discoloration..	Minimize	[12,18]
	Root rot rate	C2	Rot rate of peony root and the rate corresponding to different waterlogging tolerance is 0–30% (strong), 30–60% (medium) and 60–100% (weak).	Minimize	[19]
	Disease rate	C3	Number of diseased plants/number of healthy plants.	Minimize	[18,21]
	Survival rate	C4	Number of surviving plants after three years of cultivation/total number of plants at the beginning.	Maximize	[18,21]
Ornamental features	Flower number per plant	C5	Average number of normally open flowers.	Maximize	[18,19]
	Group blooming period	C6	The day number between the date of the first flower blooming and the date of the last flower falling of a group plants.	Maximize	[18]
	Flower type	C7	According to the shape of the petal, flowers are characterized as different types.	Valved, complex	[17,19]
	Proportion of flowering plant	C8	Number of flowering plants/total number of plants.	Maximize	[21]
	Aborted flowers per plant	C9	Aborted flowers were defined as Flowers could not open and always maintain in the small size and juvenile stage during the full-blooming period.	Minimize	[20]
	Blooming period per flower	C10	Days between the date of the petal blooming and the first petal falling.	Maximize	[18]
	Flower diameter	C11	Average flower diameter in full bloom.	Maximize	[19]
Growth habits	Stem bending degree	C12	The angle between the stem and the vertical direction of the ground.	Minimize	[20,21]
	Plant height	C13	The distance (cm) from the ground to the top of the plant.	Medium	[21]
	Plant width	C14	Maximum width (cm) of aboveground projection of the plant.	Medium	[21]
	Stem number	C15	A mature and normal stem has at least three compound leaves and the height should be more than twenty cm.	Maximize	[21]
	Stem diameter	C16	Diameter of a plant’s mature and healthy stem five centimeters above.	Maximize	[20]
	Dates of bud break	C17	Bud break is the opening of bud scales and the emergence of delicate shoots or leaves. emerged.	Earlier	[18,55]
	Chlorophyll content	C18	Average chlorophyll content of peony leaves on May 15.	Maximize	[12]

**Table 5 ijms-23-14342-t005:** Scales of pairwise comparison.

Scales	Definition	Explanation
1	Equal importance	Two activities contribute equally to the objective
3	Slight importance	Experience and judgment moderately favor one activity over another
5	Strong importance	Experience and judgement strongly favor one activity over another
7	Very strong importance	An activity is strongly favored and its dominance demonstrated in practice
9	Extreme importance	An activity is extremely favored and its dominance demonstrated in practice
2, 4, 6, 8	Intermediate values	Importance between two corresponding adjacent levels above mentioned

The importance definition between different traits is in accordance with the description proposed by Saaty (1980) [28].

**Table 6 ijms-23-14342-t006:** An example of AHP questionnaire used to determine the relative importance of a criteria.

Importance	More Important								Equal	Less Important								Criteria
	9	8	7	6	5	4	3	2	1	1/2	1/3	1/4	1/5	1/6	1/7	1/8	1/9	
Adaptability									√									Adaptability
							√											Ornamental features
								√										Growth habits
Ornamental features											√							Adaptability
									√									Ornamental features
								√										Growth habits
Growth habits											√							Adaptability
										√								Ornamental features
									√									Growth habits

The ratio of the importance is in accordance with Table 5.

**Table 7 ijms-23-14342-t007:** The pairwise comparison matrix established in this study.

	Judgment Matrix							
A-B	A	B1	B2	B3				
	B1	1	2	3				
	B2	1/2	1	2				
	B3	1/3	1/2	1				
B1-C	B1	C1	C2	C3	C4			
	C1	1	2	2	3			
	C2	1/2	1	1	2			
	C3	1/2	1	1	2			
	C4	1/3	1/2	1/2	1			
B2-C	B2	C5	C6	C7	C8	C9	C10	C11
	C5	1	1	3	2	4	2	4
	C6	1	1	3	2	4	2	4
	C7	1/3	1/3	1	1/2	2	1/2	2
	C8	1/2	1/2	2	1	2	1	2
	C9	1/4	1/4	1/2	1/2	1	2	1
	C10	1/2	1/2	2	1	1/2	1	3
	C11	1/4	1/4	1/2	1/2	1	1/3	1
B3-C	B3	C12	C13	C14	C15	C16	C17	C18
	C12	1	3	4	3	2	2	3
	C13	1/3	1	2	1	1/2	1/2	1
	C14	1/4	1/2	1	1/2	1/3	1/3	1/2
	C15	1/3	1	2	1	1/2	1/2	1
	C16	1/2	2	3	2	1	1	2
	C17	1/2	2	3	2	1	1	2
	C18	1/3	1	2	1	1/2	1/2	1

The ratio of the importance is in accordance with Table 5.

**Table 8 ijms-23-14342-t008:** Random Index (RI) values.

Number of Criteria	1	2	3	4	5	6	7	8	9	10
RI	0.0	0.0	0.52	0.89	1.11	1.25	1.35	1.40	1.45	1.49

**Table 9 ijms-23-14342-t009:** Rating scale of all sub-criteria.

Sub Criteria	Score			References
	1/3	2/3	1	
Heat damage level (C1)	4–5	3.5–4	3–3.5	[12,18]
Root rot rate (C2)	Weak	Medium	Strong	[19]
Disease rate (C3)	40–80%	20–40%	0–20%	[18,21]
Survival rate (C4)	<60%	60–80%	>80%	[18,21]
Flower number per plant (C5)	<2	2-4	>4	[18,19]
Group blooming period (C6)	<10 days	10–15 days	>15 days	[18]
Flower type (C7)	Single or lotus form	Anemone or Crown form	Proliferation form	[19]
Proportion of flowering plant (C8)	<50%	50–80%	>80%	[21]
Aborted flowers per plant (C9)	>2	1-2	<1	[20]
Blooming period per flower (C10)	<5 days	5–6 days	>6 days	[18]
Flower diameter (C11)	<11 cm	11–13 cm	>13 cm	[19]
Stem bending degree (C12)	>30 °C	10–30 °C	<10 °C	[20,21]
Plant height (C13)	>60 or <40 cm	50–60 cm	40–50 cm	[21]
Plant width (C14)	>60 or <40 cm	50–60 cm	40–50 cm	[21]
Stem number (C15)	<4	4-5	>5	[21]
Stem diameter (C16)	<5 mm	5–6 mm	>6 mm	[20]
Dates of bud break (C17)	Later than March 15	March 1–March 15	Earlier than March 1	[18]
Chlorophyll content (C18)	<50	50-55	>55	[12]

## Data Availability

The data that support the findings of this study are available from the corresponding author, upon reasonable request.

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
