# Peer review of "Development of a Multi-Criteria Decision-Making Approach for Evaluating the Comprehensive Application of Herbaceous Peony at Low Latitudes"

_ijms, 2022, doi:10.3390/ijms232214342_

Round 1

Reviewer 1 Report

At page 348, fertilization may be replaced by fertilizer application

At page 372, were the found should be were found the

Author Response

We appreciated for your warm work earnestly‚ and we have modified these two points as suggested.

Reviewer 2 Report

-The authors need to add data that give or show the importance of this herbaceous peony, for example, economic, social, and Ambiental data.

-In lines 78 and 79, the authors refer to high temperatures and humidity but do not mention numerical data.

- In Table 1. “Rot rate of peony root and the rate corresponding to different waterlogging tolerance”. Clarify if the rate describes the root or waterlogging tolerance. In the case of waterlogging tolerance, the objective is to maximize, but if you refer to the root, it is minimize; it is unclear in the table.

-In Line 141 is a missing space between Table 3 and the subsequent text.

-In lines 153 to 163, the description of the results in Figure 3 could be expanded further.

-In line 185, it is recommended to review the description of the results in section 2.5, “Observations of growth habits-related traits,” because not only 'Taohua Feixue' and 'Yangfei Chuyu' have limited height and width, but also Zhuguang.

- In lines 186 to 188. The authors could review the description of figures 5 A and B results because some cultivars suffer changes from 2018 to 2021 and are not mentioned.

-     In line 204. Change Tables 4 for Table 4.

-      In lines 207 to 216, the authors could avoid using “Table 4” many times for the same paragraph.

-      In line 223, the meaning of the abbreviation of ETR does not appear where it is first mentioned.

-   In line 334. Change on the previous study for previous studies.

-   Add the letter N to the picture in figure 7.

-    In lines 372 to 374. High temperature and high humidity were the found most important factors restricting the popularization and application of peony under relatively high temperatures at low latitudes refers to the result section not to methodology section; it is recommended to send it to the results section.

-In line 380, the word “recommendation” should be in plural “recommendations”.

-In line 395, remove the point after Equation 1.

-Line 437, 440 and 441. There is not clear why measure chlorophyll is only in three cultivars and not in all.

-The authors mention two growing seasons (2018-2021) and that four-year-old peony tuberous roots were used in this study. Please clarify how long it would take to test new cultivars with the MCDM model.

-If possible, briefly mention the benefit of using the MCDM model vs not using the model (economic, time, surface required for testing, etc).

-The authors mention that the objects of this research were to 1.- develop an MCDM model for evaluating the comprehensive application of herbaceous peony at low latitudes and 2.- select representative peony cultivars with strong adaptability for crossbreeding and studies on the mechanism of stress resistance. Please explain what you mean by “mechanism of stress resistance” and how you achieved this object with the present study.

-There are errors in lines 142, C4 instead C5, and C5 instead C6.

-It is unclear how table 4 is obtained; the authors need to explain deeper.

-In line 264 and 265, why do the authors establish the heat damage level, waterlogging resistance, and disease rate as the three evaluation criteria of the MCDM model?

-To employ this model in other plant species, what could be the procedure to use it?

-In lines 347 and 348, the authors need to give more details about normal field management and fertilization of the cultivars employed in this work.

-In figure 9, how was the procedure of literature and investigation? It is supposed that surveys are part of the investigation; the authors mentioned some related to surveys, how are the surveys included in the model?

-Of figure 9, details about Building decision hierarchy systems are missing

-In figure 9, details about calculating normalized weight are missing

-In figure 9, details about calculating consistency are missing

-Of figure 9, it is not clear how the computation of global weight is achieved

-In figure 9, how the field observation and data collection is written after weighting the criteria indices? Does it mean that weighting the criteria indices does not include field observation and data collection? The authors need to be more explicit.

-In line 377, the authors mentioned that the judgment of experts involved facilitates the weight criteria; how was this done? How many experts were surveyed, what question were asked and how the results of these surveys were included in the model. The authors need to give a detailed explanation.

-In the square matrix of pair-wise comparison, it is not clear how the ratio of the importance of i-th trait compared with j-th trait is calculated?. The authors mentioned table 5; however, the authors need to give more details about the meaning of slightly, strongly, extremely, and equal.

-It is unclear how RI is obtained or what equation was used to calculate this.

-From the fifteen species, how many plants of each specie were monitored?

Reviewer 3 Report

The paper with the title “Development of a multi-criteria decision-making approach for evaluating the comprehensive application of herbaceous peony at low latitudes” used multi-criteria decision-making for fifteen representative herbaceous peony cultivars. The model proposed provides a reference for the large-scale introduction, breeding and application of adapted herbaceous peony cultivars under ever-changing unfavorable climatic conditions.

Abstract
Contains some unnecessary details but lacks clear delimitation or highlight of the novelty. For example, you could say “This paper proposes an original model for assessing… ”. The model was validated on 15 herbaceous peonies cultivars from … Also, consider the possibility that your model might become useful for assessing genotypes of other species beyond peonies, by other authors and you could say “the model could be adapted for other species…”.

Introduction
Is comprehensive enough but can be improved. At line 51 it starts presenting MCDM. Before this paragraph I advise authors to introduce a paragraph in which they explain that currently there is no harmonized methodology for what they want to study, therefore there is needed a model for this.  

Results
Some parts of the results section seem to be a methodology, therefore should be part of Material and Method Section. I refer here subchapters 2.1 and 2.2.

Based on a paragraph from lines 50-70, and material and method + results => the manuscript does not outline sufficiently, or does not succeed to provide clear delimitation in terms of the novelty part regarding the model. Is the model proposed entirely original? Then, authors should say so. Alternatively, is an extension of an existing model? Please use a clear terminology both in the abstract and discussion section, delimiting original contribution to the model and the existing literature, explaining what this particular model has to offer and how can be further adapted.

The Discussion section ends up abruptly. I propose authors to insert a subchapter 3.5. Future perspectives. In this chapter, you might explain to readers how they can use this model in their research, or suggest how they could successfully adapt it for their needs. What are the possible future applications? How can this solve some challenges? A harmonized approach might be provide a global overview and comparable data for meta-analysis directed towards pattern identification across geographical regions affected by climate change… etc.

Best regards.

Author Response

Response to Reviewer 3

Comments and Suggestions for Authors

The paper with the title “Development of a multi-criteria decision-making approach for evaluating the comprehensive application of herbaceous peony at low latitudes” used multi-criteria decision-making for fifteen representative herbaceous peony cultivars. The model proposed provides a reference for the large-scale introduction, breeding and application of adapted herbaceous peony cultivars under ever-changing unfavorable climatic conditions.

Response:

We are truly grateful for Reviewer 3’s professional review work on our article. These comments are all valuable and helpful for improving our article. According to these comments, we have made extensive modifications to our manuscript, and hope that the correction will meet with approval. Once again, thank you very much for your comments and suggestions.

Comment 1

Abstract: Contains some unnecessary details but lacks clear delimitation or highlight of the novelty. For example, you could say “This paper proposes an original model for assessing… ”. The model was validated on 15 herbaceous peonies cultivars from … Also, consider the possibility that your model might become useful for assessing genotypes of other species beyond peonies, by other authors and you could say “the model could be adapted for other species…”.

Response: Thanks to the suggestions, we have simplified the description of unnecessary details and added features and advantages about the model in the abstract.

Comment 2

Introduction: Is comprehensive enough but can be improved. At line 51 it starts presenting MCDM. Before this paragraph I advise authors to introduce a paragraph in which they explain that currently there is no harmonized methodology for what they want to study, therefore there is needed a model for this. 

Response: Thank you for pointing this out. We agree with the reviewer that before starts presenting MCDM we should point out the context in which such a model is needed. We have made a lot of supplements to the introduction, especially the current application status of peony at low latitudes and also a paragraph in which explains that currently there is no harmonized methodology for what we want to study.

Comment 3

Results: Some parts of the results section seem to be a methodology, therefore should be part of Material and Method Section. I refer here subchapters 2.1 and 2.2.

Response: We agree with the reviewer’s assessment. Accordingly, we made a major restructuring of the article, adjusting subchapters 2.1 and part of the 2.2 to the Method Section. However, local and global weights of the criterias still keep in the results section, mainly because this part is the result of our model building and represents the specificity of the model.

Comment 4

Based on a paragraph from lines 50-70, and material and method + results => the manuscript does not outline sufficiently, or does not succeed to provide clear delimitation in terms of the novelty part regarding the model. Is the model proposed entirely original? Then, authors should say so. Alternatively, is an extension of an existing model? Please use a clear terminology both in the abstract and discussion section, delimiting original contribution to the model and the existing literature, explaining what this particular model has to offer and how can be further adapted.

Response: We greatly appreciate the constructive suggestions, and we did neglect to introduce the novelty of this model. Exactly, this study creates a multi-criteria integrated decision support framework for selecting elite herbaceous peony germplasm at low latitudes. The MCDM model established in this study is an extension and improvement of our previous study, aiming to evaluate the comprehensive application of herbaceous peony at low latitudes. The model reconstructed the AHP system, increased adaptability and growth habits-related indices, while reduced reproductive traits and ornamental values-related indices. In addition, the weight of adaptability-related indices was improved via a pair-wise comparison. We have added this description in the introduction in the revised manuscript. In addition, a discussion of the advantages and disadvantages of the model was also added in the Discussion section.

Comment 5

The Discussion section ends up abruptly. I propose authors to insert a subchapter 3.5. Future perspectives. In this chapter, you might explain to readers how they can use this model in their research, or suggest how they could successfully adapt it for their needs. What are the possible future applications? How can this solve some challenges? A harmonized approach might be provide a global overview and comparable data for meta-analysis directed towards pattern identification across geographical regions affected by climate change… etc.

Response: Based on the above valuable suggestions, we have inserted a subchapter 3.5: Limitations, recommendations and future perspectives to the Discussion section. In this section, we discussed all the aspects mentioned by the reviewer and gave an outlook on future applications.

Reviewer 4 Report

The present paper proposes a multi-criteria decision-making (MCDM) model for selecting peonies adapted to low-latitude climate. The topic presented in this work is really interesting. However several challenges are required:

I analyze the single sections:

Abstract has inappropriate structure. I suggest to answer the following aspects: - general context - novelty of the work - methodology used (describe briefly the main methods or treatments applied) - main results and related interpretations.

Introduction: This section should briefly place the study in a wide context and emphasize why it is relevant carrying out the analysis. It should define the purpose of the work and its significance. In this perspective, this section is too succinct and fails to effectively point out the relevance of your contribution towards the existing literature. Moreover, the authors do not provide at the end of the section the description of the paper structure which is very useful for readers.

I would suggest the authors to expand this section broadly focusing on literature applying MCDA. Please see.

https://doi.org/10.1016/j.ecolecon.2020.106794

https://doi.org/10.1016/j.enpol.2019.111220

https://doi.org/10.3390/su141811276 

 Materials and methods: I found this section very important for the readability of the paper. However, several challenges need to be addressed. Methods should be described in detail. I think the research procedure could be much more clearly described by means of a diagram also highlighting its potential and limit.

Discussions: The discussion of the results is merely descriptive and the obtained evidence is flimsy due to the fact the outcomes are not supported by an adequate discussion in light of scientific literature. Authors should discuss the results and how they can be interpreted in perspective of previous studies and their implications should be discussed in the broadest context possible.

Conclusions: Conclusions must also be revised according to the previous comments. In particular, they should discuss practical and policy implications as well as future lines of research.

Author Response

Response to Reviewer 4

Comments and Suggestions for Authors

The present paper proposes a multi-criteria decision-making (MCDM) model for selecting peonies adapted to low-latitude climate. The topic presented in this work is really interesting. However several challenges are required:

Response: We are truly grateful for Reviewer 4’s professional review work on our article. We have tried our best to revise the manuscript and we sincerely hope that this revised manuscript has addressed all your comments and suggestions. Once again, thank you very much for your comments and suggestions.

Comment 1 

Abstract has inappropriate structure. I suggest to answer the following aspects: - general context - novelty of the work - methodology used (describe briefly the main methods or treatments applied) - main results and related interpretations.

Response: Thanks to the suggestions, the abstract has been modified according to the principles and aspects you mentioned. Moreover, we have simplified the description of unnecessary details of the main methods and added novelty of the work in the abstract.

Comment 2

Introduction: This section should briefly place the study in a wide context and emphasize why it is relevant carrying out the analysis. It should define the purpose of the work and its significance. In this perspective, this section is too succinct and fails to effectively point out the relevance of your contribution towards the existing literature. Moreover, the authors do not provide at the end of the section the description of the paper structure which is very useful for readers.

I would suggest the authors to expand this section broadly focusing on literature applying MCDA. Please see.

https://doi.org/10.1016/j.ecolecon.2020.106794

https://doi.org/10.1016/j.enpol.2019.111220

https://doi.org/10.3390/su141811276 

Response: We greatly appreciate the constructive suggestions. We have made a lot of supplements to the introduction, especially the current application status of peony at low latitudes and also a paragraph (in lines 87-102) in which explain that currently there is no harmonized methodology for what we want to study. In addition, at the end of the Introduction section, a paragraph (in lines 167-170) in which described the paper structure was provided based on the literature applying MCDA.

Comment 3

Materials and methods: I found this section very important for the readability of the paper. However, several challenges need to be addressed. Methods should be described in detail. I think the research procedure could be much more clearly described by means of a diagram also highlighting its potential and limit.

Response: We agree with the reviewer’s comment and have made extensive modifications to this section. Firstly, we restructured the section by putting the plant material and cultivation conditions at the end as a case study and adjusting subchapters 2.1 and part of the 2.2 to the Method section, and secondly, we supplemented a lot of details about the model building by adding five tables and related descriptions to make the section more understandable.

Comment 4

Discussions: The discussion of the results is merely descriptive and the obtained evidence is flimsy due to the fact the outcomes are not supported by an adequate discussion in light of scientific literature. Authors should discuss the results and how they can be interpreted in perspective of previous studies and their implications should be discussed in the broadest context possible.

Response: As suggested by the reviewer, the discussion section was further enriched and discussed a broader context as much as possible. Moreover, we have inserted a subchapter 3.5: Limitations, recommendations and future perspectives to the Discussion section to make the discussion more adequate.

Comment 5

Conclusions: Conclusions must also be revised according to the previous comments. In particular, they should discuss practical and policy implications as well as future lines of research.

Response: Our conclusions were incorporated in the Discussion, especially in the added subchapter 3.5: Limitations, recommendations and future perspectives, which fully summarized the strengths and weaknesses of the MCDM model and provided an outlook on its future application.

Round 2

Reviewer 3 Report

Authors have significantly improved their manuscript. All comments were addressed.

Author Response

Thank you, we appreciate your time and efforts in our manuscript.

Reviewer 4 Report

The authors did not provide the required modifications nor adeguate explanations in the rebuttal. This especially true for the introduction, methods and conclusions. I do not recommend it for publication

Author Response

We have made extensive modifications to our manuscript according to the Reviewer 4's comments in Round 1, but Reviewer 4 still gave us a rejection on the manuscript, which is so regrettable. The Reviewer seems to have missed some original and revised content of the manuscript. Nevertheless, we still made more in-depth revisions to the introduction and methods, and added a conclusion section in Round 2. Despite the negative decision, we are grateful for Reviewer 4’s review work on our article, some of the comments did improve the quality of the manuscript.

Round 3

Reviewer 4 Report

.